# Association of Primary Sjögren’s Syndrome and Vitamin B12 Deficiency: A Cross-Sectional Case-Control Study

**DOI:** 10.3390/jcm9124063

**Published:** 2020-12-16

**Authors:** Geoffrey Urbanski, Floris Chabrun, Baudouin Schaepelynck, Morgane May, Marianne Loiseau, Esther Schlumberger, Estelle Delattre, Christian Lavigne, Valentin Lacombe

**Affiliations:** 1Department of Internal Medicine, Angers University Hospital, 4 Rue Larrey, 49000 Angers, France; baudouin.schaepelynck@gmail.com (B.S.); morgane.may@aol.com (M.M.); marianne.loiseau@laposte.net (M.L.); esther.schlum@orange.fr (E.S.); Estelle.Delattre@chu-angers.fr (E.D.); ChLavigne@chu-angers.fr (C.L.); Valentin.Lacombe@chu-angers.fr (V.L.); 2Department of Biochemistry and Genetics, Angers University Hospital, 4 Rue Larrey, 49000 Angers, France; Floris.Chabrun@chu-angers.fr

**Keywords:** vitamin B12, Sjögren’s syndrome, vitamin B12 deficiency, haptocorrin

## Abstract

Descriptive and retrospective studies without control groups have suggested a possible association between primary Sjögren’s syndrome (pSS) and vitamin B12 (B12) deficiency. This is of importance because several mucosal and neurological features are common to these two conditions and could be prevented or reversed in case of B12 deficiency. We aimed to evaluate the association between pSS and B12 deficiency. We prospectively assessed the B12 status of 490 patients hospitalized in an internal medicine department over a 15-week period. Patients with pernicious anemia were excluded. We extracted patients with pSS and paired them with controls according to age and sex, with a 1:5 ratio. Twenty-one pSS patients were paired with 105 control patients. The median age was 70 years old (51–75) and 95.2% of patients were women. The plasma B12 level was lower in pSS patients (329 (293–521) ng/L vs. 456 (341–587) ng/L, *p* < 0.0001). B12 deficiency was associated with pSS (42.9% among pSS patients vs. 11.4% among controls), even after adjustment for other causes of B12 deficiency (OR 6.45 (95%CI: 2.08–20.0)). In conclusion, pSS appeared to be associated with B12 deficiency, even after the exclusion of pernicious anemia. This justifies screening and treating B12 deficiency in pSS patients.

## 1. Introduction

Vitamin B12 (B12) or cobalamin deficiency affects around 1.5% of the general population and up to 10–15% of subjects over 60 years old [1,2]. Early diagnosis and treatment of B12 deficiencies allows hematological and mucosal consequences to be corrected and irreversible neurological complications of the B12 deficiency to be avoided [1].

Various conditions are known to favor a B12 deficiency leading to the assessment of B12 status in such cases: pernicious anemia, vegan diet, and food-cobalamin malabsorption, including proton pump inhibitor or metformin treatment, for example [3]. However, about 10% of B12 deficiencies remain unexplained or are considered idiopathic [4,5], raising the possibility of other unknown causes.

The association between primary Sjögren’s syndrome (pSS) and B12 deficiency has been questioned following observations of patients with these two conditions but no other causes of B12 deficiency [6]. This possible association between pSS and B12 deficiency is of importance because several symptoms (such as chronic fatigue, neuropathy, or mucosal features) are common to these two conditions and could be prevented or reversed in cases of B12 deficiency [7,8]. Lundström et al. and Andrès et al. reported quite a high frequency of B12 deficiency in pSS in two descriptive studies, 13% and 8.8%, respectively [9,10]. These results raised the importance of screening this deficiency among pSS patients, but, in the absence of a control group, the authors cannot reach conclusions about the association between these two conditions. Moreover, these studies defined B12 deficiency by using solely the plasma B12 measurement, whereas experts now recommend using different biomarkers to assess B12 status [11,12,13].

We aimed to assess the association between pSS and B12 deficiency in a cross-sectional case-control study after adjustment for other causes of B12 deficiency.

## 2. Methods

### 2.1. Ethics

This study was approved by the Bioethics Committee of Angers University Hospital (no. 2015-15). We applied the Strengthening the Reporting of Observational Studies in Epidemiology (STROBE) statement to observational studies. All the patients gave informed consent.

### 2.2. Study Population

We prospectively investigated the B12 status of all in-patients hospitalized in the department of internal medicine, Angers University Hospital, with the exception of patients who did not require blood samples. Recruitment lasted for 15 weeks. From this cohort, we extracted patients with pSS and control patients paired by sex and age (±3 years), with a 1:5 ratio. The patients in the control group were randomly selected from the subjects without pSS that met the matching criteria.

Patients receiving vitamin B12 supplementation, with a genetic cause for B12 deficiency, with systemic immune disease (except for pSS), and with pernicious anemia were excluded. Gastric parietal cells and intrinsic factor autoantibodies were systematically searched for patients presenting with B12 deficiency in order to exclude pernicious anemia.

### 2.3. Definitions of Primary Sjögren’s Syndrome and Nutrient Deficiencies

pSS was defined according to the ACR/EULAR 2016 criteria [14].

B12 deficiency was defined by a total plasma B12 < 200 ng/L or by the association of a total plasma B12 between 201–350 ng/L and an elevation of: (i) plasma homocysteine (≥13 µmol/L for women, ≥15 µmol/L for men, ≥20 µmol/L for patients older than 65 years old, but not considered in case of plasma folate <4 µg/L [15,16]); and/or (ii) plasma methylmalonic acid (≥0.35 µmol/L [17]), as recommended by experts [12].

Folate deficiency was defined by a level of plasma folate <4 µg/L [18].

Iron deficiency was defined as follows: (i) a level of serum ferritin ≤30 µg/L; or (ii) a level of serum ferritin ≤100 µg/L in patients with C-reactive protein ≥10 mg/L [19].

### 2.4. Data Collection

All patients were investigated for B12 status. Measurements of plasma B12, plasma folate, and ferritin were systematically performed, and measurements of the other biomarkers of B12 deficiency (plasma homocysteine, plasma methylmalonic acid) were performed in case of plasma B12 between 201 and 350 ng/L. Indeed, where plasma B12 is in the low normal range, B12 deficiency must be assessed by measurements of other biomarkers because measuring plasma vitamin B12 alone is not enough to identify B12 deficiency [11,20,21].

The total plasma vitamin B12 was determined by competitive chemiluminescent immunoassay on ADVIA Centaur^®^ system (Siemens Healthcare Diagnostics Inc., Tarrytown, NY 10591-5097, USA). The plasma homocysteine was determined by HPLC-MS/MS (Agilent 1200 Infinity Series, Agilent Technologies, Santa Clara, CA, USA; Triple Quad™ 4500, SCIEX, Framingham, MA, USA). The plasma methylmalonic acid was determined by HPLC-MS/MS (Agilent 1200 Infinity Series, Agilent Technologies, Santa Clara, CA, USA; Triple Quad™ 5500, SCIEX, Framingham, MA, USA).

The data for potential causes of B12 deficiency were collected: chronic use of metformin or antacids, *Helicobacter pylori* infection, veganism, and fundic gastric or ileum resection. Metformin or antacid intake was defined as chronic in case of use >6 months. *Helicobacter pylori* infection was defined by a positive ^13^C urea breath test.

### 2.5. Statistical Analysis

The quantitative data are presented in medians and quartiles, and compared using the sign test. The qualitative data are presented as absolute values and as percentages and were compared using the marginal homogeneity test. The multivariable analysis for causes of B12 deficiency was performed with binary logistic regression. The alpha risk was 5%. The odds ratios (OR) were presented with a confidence interval of 95%. The analyses were carried out using SPSS software v23.0 (IBM Corp, New York, NY, USA).

## 3. Results

During the study period, 490 patients were assessed for B12 status, including 21 patients with pSS. No patients presented with a combination of pSS and pernicious anemia. The 21 pSS patients were paired for age and sex with 105 controls. Characteristics of the study population are described in Table 1.

The plasma B12 level was lower in pSS patients than in the control patients (*p* < 0.0001, Figure 1 and Table 2). A B12 deficiency was observed in 9/21 (42.9%) pSS patients and 12/105 control patients (11.4%, *p* < 0.0001). Iron deficiency was more frequent in pSS patients (5/21, 23.8%) than in the control group (13/105, 12.4%, *p* = 0.02). The frequency of folate deficiency did not differ between cases and controls (*p* = 0.86, Table 2).

After adjustment for age and other causes for B12 deficiency (chronic metformin and antacid use and *Helicobacter pylori* infection), only pSS was significantly associated with B12 deficiency with an OR of 6.45 [95%CI: 2.08–20.0] (*p* < 0.001) (age: OR 1.001 [95%CI: 0.97–1.033], metformin use: OR 5.08 [95%CI: 0.48–53.4], antacid use: OR 2.58 [95%CI: 0.83–8.0], *Helicobacter pylori* infection: OR 3.54 [95%CI: 0.57–21.8]).

Among pSS patients, patients with B12 deficiency tended to be younger (57 (49–77) vs. 71.5 (60.8–75) years old in pSS patients without B12 deficiency), to present more frequently with an iron deficiency (44.4% vs. 16.7%), and to have lower levels of hemoglobin (121 (113–137) vs. 135 (129–142) g/L, Appendix A).

## 4. Discussion

The association between pSS and B12 deficiency has been suggested in previous case series and descriptive studies [6,9,10]. To our knowledge, our study is the first to prospectively assess the frequency of B12 deficiency in pSS and the first to compare this prevalence to a control group. We demonstrated a strong association between pSS and B12 deficiency in pSS compared to paired controls (OR 6.45 (95%CI: 2.08–20.0)). As 42.9% of pSS in-patients presented a B12 deficiency, our results highlighted the interest of screening B12 deficiency in cases of pSS.

In the control group of this study, the prevalence of B12 deficiency was 11.4%, which is in line with previous studies about patients over 60 years of age [2]. Most B12 deficiencies in our study were not responsible for hematological impairment. This may be explained by the study design with systematic screening for B12 deficiency.

pSS can be associated with other immune diseases, notably pernicious anemia [22,23]. In the study conducted by Lundström et al., which suggested that pSS could cause B12 deficiency, pernicious anemia was diagnosed in most pSS patients with B12 deficiency [9]. As pernicious anemia is one of the main causes for B12 deficiency, we systematically searched for it in patients with B12 deficiency in the study and excluded them from this analysis. As a consequence, the association between pSS and B12 deficiency that we demonstrated is not mediated by pernicious anemia.

Different physiological mechanisms may explain how pSS could lead to B12 deficiency [22]. We demonstrated that pSS was also associated with iron deficiency, notably in patients with B12 deficiency. This could well present evidence of a common mechanism shared by these two deficiencies. A loss of gastric acidity caused by lymphocytic gastric infiltration might explain both iron and B12 deficiencies, but the level of gastrin (raised in the case of gastric hypochlorhydria) did not differ between pSS and control patients in our study. Otherwise, an altered production of salivary carrier protein could explain those deficiencies in the absence of elevated gastrin levels in pSS patients. In particular, a quantitative or qualitative salivary haptocorrin deficiency may explain the B12 deficiency [1].

Our study has a few limitations. We only included hospitalized pSS patients, whose characteristics may differ from outpatients with pSS. It could be interesting to study B12 deficiency in pSS outpatients. However, the majority of pSS patients included in this study were hospitalized for planned explorations following the pSS diagnosis. Thus, they are reasonably similar to outpatients. One could notice that they were older compared to cohort studies and that B12 deficiency is more frequent in older people. However, we noticed that pSS patients with B12 deficiency in our study were younger than pSS patients without B12 deficiency and of similar age than patients described in pSS cohorts from the literature [24,25]. The proportion of pSS patients without anti-SSA antibodies is higher in our population than in previously described large cohorts [26]. The first explanation could be that patients were recruited in an Internal Medicine department, thus, characteristics of patients may differ from studies conducted in Rheumatology departments. Additionally, we systematically performed minor salivary gland biopsy in the case of suspected pSS at the same time than the search for anti-SSA antibodies, which allows to avoid underdiagnosis in patients without anti-SSA antibodies.

The number of pSS patients in our study did not allow us to determine clinical differences according to B12 status. We may hypothesize that some clinical pSS manifestations could be more frequent in pSS patients with B12 deficiency. Indeed, as both pSS and B12 deficiency may cause neuropathy, we may speculate that pSS patients with B12 deficiency could be more prone to suffer from small or large fiber neuropathy [1,8].

## 5. Conclusions

We demonstrated that pSS was independently associated with B12 deficiency, even after excluding associated pernicious anemia, with six times more B12 deficiencies in pSS patients than in controls. This justifies screening and treating B12 deficiency in pSS patients.

## Figures and Tables

**Figure 1 jcm-09-04063-f001:**
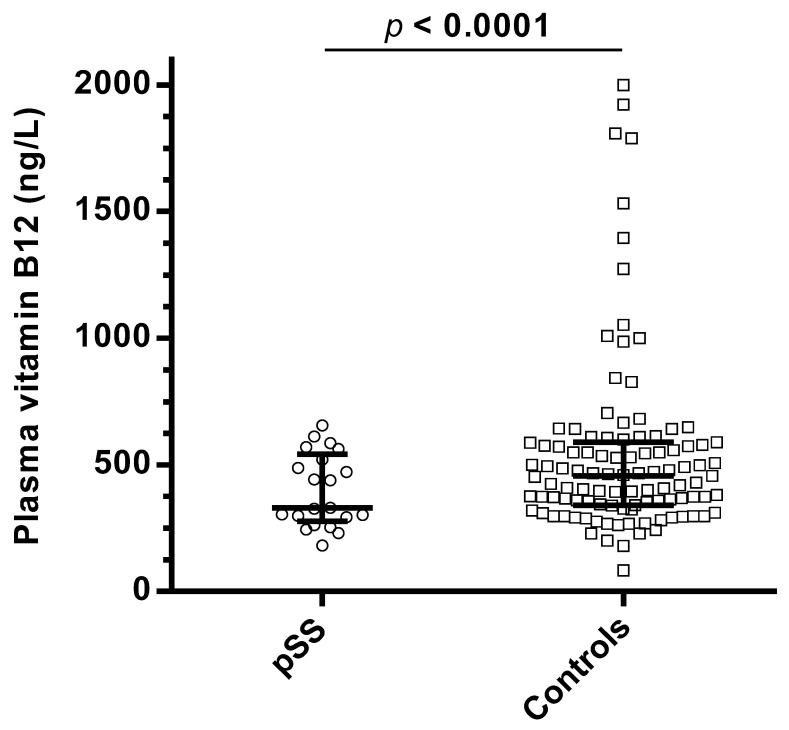
Distribution of vitamin B12 levels in pSS patients and controls. Notes: Plasma vitamin B12 is presented with median and quartiles.

**Table 1 jcm-09-04063-t001:** Description of the study population.

	pSS Patients	Controls
Number of patients	21	105
Age (years)	70 (51–75)	70 (51–77)
Women	20 (95.2%)	100 (95.2%)
Body mass index (kg/m²)	25.5 (22.3–29.4)	25.9 (21.8–30.4)
pSS duration at time of B12 measurement (months)	12 (2–46)	
Schirmer’s test ≤ 5 mm/5 min	13 (61.9%)	-
Unstimulated whole saliva flow ≤1.5 mL/15 min	12 (57.1%)	-
Lymphocytic sialadenitis with focus score ≥ 1	20 (95.2%)	-
Anti-SSA antibodies	7 (33.3%)	-
Anti-SSB antibodies	5 (23.8%)	-
ANA positivity	19 (90.5%)	-
ANA ≥ 1/320	14 (66.7%)	-
Rheumatoid factor	12 (57.1%)	-
Age at pSS diagnosis (years)	64 (49–71)	-
Extra-glandular manifestations	15 (71.4%)	-
Inflammatory arthralgia	7 (33.3%)	-
Polyarthritis	1 (4.8%)	-
Parotidomegaly	4 (19.0%)	-
Serositis	1 (4.8%)	-
Large fiber neuropathy	3 (14.3%)	-
Small fiber neuropathy	2 (9.5%)	-
CNS involvement *	2 (9.5%)	-
Pulmonary interstitial disease	2 (9.5%)	-
Lymphoma	1 (4.8%)	-

ANA: Anti-nuclear antibodies; CNS: Central nervous system; pSS: primary Sjögren’s syndrome. * One patient presented mild cognitive troubles and many hypersignals on RMN in the white matter; the second patient had a demyelinizating CNS disease without fulfilling the criteria for multiple sclerosis.

**Table 2 jcm-09-04063-t002:** Comparison of nutrient characteristics in the two groups.

	pSS	Controls	*p* Value
Number of patients	21	105	
Vitamin B12 status			
Plasma vitamin B12 (ng/L, NR 350–1000)	329 (293–521)	456 (341–587)	<0.0001
Plasma homocysteine (µmol/L *)	17.6 (13.9–19.4)	14.3 (10.3–17.9)	0.07
Plasma methylmalonic acid (µmol/L, NR ≤ 0.35)	0.48 (0.20–0.56)	0.31 (0.20–0.45)	0.66
Number of patients with B12 deficiency	9 (42.9%)	12 (11.4%)	<0.0001
Other biological measurements			
Serum folate (µg/L, NR ≥ 4)	5.2 (4.3–7.5)	6.2 (4.4–8.3)	0.24
Folate deficiency (<4 µg/L)	3 (14.3%)	16 (15.2%)	0.86
Serum ferritin level (µg/L, NR > 30)	99 (47–151)	142 (83–252)	<0.0001
Iron deficiency	5 (23.8%)	13 (12.4%)	0.02
Creatinine clearance (mL/min/1.73 m²) MDRD (NR ≥ 90)	89.5 (69.2–100.6)	85.3 (65.2–102.8)	0.77
Serum gastrin (pg/mL, NR 28–115)	46.3 (38.1–66.1)	55.1 (44.2–71.3)	0.36
Other causes of B12 deficiency among patients with B12 deficiency			
Metformin use	0/9 (0%)	2/12 (16.7%)	0.04
Chronic antacid use	5/9 (55.6%)	3/12 (25.0%)	0.71
Nutritional deficiency	0/9 (0%)	0/12 (0%)	>0.99
Fundic gastric or ileum resection	1/9 (11.1%)	1/12 (8.3%)	0.10
*Helicobacter pylori* gastric infection	0/9 (0%)	2/12 (16.7%)	0.04

Notes: NR: Normal range; pSS: primary Sjögren’s syndrome. * Normal values of homocysteine were <13 µmol/L for women, <15µmol/L for men, and <20 µmol/L for patients older than 65 years.

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
