# Peer review of "Association of Primary Sjögren’s Syndrome and Vitamin B12 Deficiency: A Cross-Sectional Case-Control Study"

_jcm, 2020, doi:10.3390/jcm9124063_

Round 1

Reviewer 1 Report

Urbanski et al investigate the association of vitamin B12 deficiency with Sjögren’s syndrome (SS) in a prospective study and in accordance with previous studies they found that SS patients have lower levels of plasma vitamin B12 compared to controls. The novelty of this study is that it has been performed prospectively in hospitalized patients (SS and non-SS controls), whereas the major disadvantage that includes small number of SS patients (n=21), which does not permit the examination of correlations with certain disease features.

Comments

  1. The study does not provide evidence that SS causes/leads vitamin B12 deficiency, only that it associates with lower plasma vitamin B12 levels. The authors do not have counts before disease onset that is needed for such a claim. Thus, such strong statements should be replaced throughout the manuscript from other, more mild/accurate, such as association, correlation etc.
  2. Disease duration should be added in Table 1 and supplementary Table.
  3. The presentation of vitamin B12 levels in SS patients and controls would help in the readily evaluation of its distribution in the studied groups.

Reviewer 2 Report

In this study the authors have measured vitamin B12 levels in 490 hospitalized patients and identified 21 patients with pSS, who were matched to 105 control patients. Patients with pSS had significantly lower B12 levels and a higher frequency of B12 deficiency according to definitions. The approach is interesting, however there are several concerns.

The title and lines 134, 165 are misleading, this is a cross-sectional case-control study and not prospective. I understand B12 was only measured once and not followed up.

It is unclear why these patients were hospitalized? PSS per se is seldom a cause for hospitalization. The pSS patients must have had concomitant conditions for which they were hospitalized. This must be stated, preferably listed in a table and discussed. Could other conditions have contributed to the B12 deficiencies? Was there a difference between causes for hospitalization between pSS and controls?

The pSS was defined according to ACR/EULAR 2016 critera. However, the frequencies of a pathological Schirmer´s test and UWS are low (around 60 %). Commonly this is around 75% in a pSS cohort. Even more conspicuous is the low frequency of SSA antibodies 33%, commonly 70-75% in a pSS cohort (ref. Brito-Zerón P, et al. Clin Exp Rheumatol. 2018). If patients are SSA negative, a positive biopsy is required. I can see 95% had a focus score ≥1 and thus these pSS seem to fullfil criteria. However, a FS of 1 in SSA negative pSS is not always convincing for autoimmunity. I somewhat question the pSS diagnosis, when so few are anti-SSA positive.

Extraglandular manifestations are less common in anti-SSA negative pSS. Which CNS manifestations did occur? Were the manifestations listed in Table 1, recorded in the patient charts or were any investigations performed for this study? Was the original pSS diagnosis made by a Rheumatologist? How many were anti-SSB positive? ANA positive?

Line 95 and table 2 “antacids”, in table S1 “proton-pump inhibitor”. Which is it?

Table 2 header is missing. Add reference values.

Creatinine clearance was calculated using the MDRD formula (sex, age, creatinine, race optional). Normal range is 65-115. The median is 120 for pSS. This seems very high for a group of patients being 70 years old. Providing a female, 70 years, white, the creatinine must be 47 to yield a GFR of 120. Were the pSS patients muscle wasted, or has any miscalculation occurred?

Table S1: Regimen deficiency, what is that?

Patients with pSS and B12 deficiency were younger than pSS without, and had more iron deficiency. Where there any vegans/vegetarians, or was dietary intake adjusted for? Table 2 Nutritional deficiency, how was that defined/assessed?

The conclusion in the abstract and text “pSS cause of B12 deficiency”, is not mechanistically supported in this study. "Associated with" is correct.

Typo

Table 1 seritis, is serositis

Round 2

Reviewer 2 Report

Thank you for responding to my questions and concerns. I still have some concerns about the pSS group itself, being different from usual pSS cohorts regarding SSA-Ab and UWS. I appreciate all pSS fulfil criteria. It is unclear how to which extent the results from this study can be extrapolated to a larger pSS cohort diagnosed within Rheumatology.

However, the paper now explains the inclusion procedure and discusses this issue. The paper has improved and the results are now clearly presented and errors corrected.